# RETRO: REUSING TEACHER PROJECTION HEAD FOR EFFICIENT EMBEDDING DISTILLATION ON LIGHTWEIGHT MODELS VIA SELF-SUPERVISED LEARNING

## ABSTRACT

Self-supervised learning (SSL) is gaining attention for its ability to learn effective representations with large amounts of unlabeled data. Lightweight models can be distilled from larger self-supervised pre-trained models using contrastive and consistency constraints, but the different sizes of the projection heads make it challenging for students to accurately mimic the teacher's embedding. We propose RETRO, which reuses the teacher's projection head for students, and our experimental results demonstrate significant improvements over the state-of-the-art on all lightweight models. For instance, when training EfficientNet-B0 using ResNet-50/101/152 as teachers, our approach improves the linear result on ImageNet to 66.9%, 69.3%, and 69.8%, respectively, with significantly fewer parameters.

## 1 INTRODUCTION

Deep learning has achieved remarkable success in various visual tasks, such as image classification, object detection, and semantic segmentation, thanks to the availability of large-scale annotated datasets. However, acquiring labeled data is time-consuming and expensive, making it crucial to explore better ways to utilize unlabeled data. Self-supervised learning (SSL) has emerged as an effective method to learn useful representations on unlabeled data, resulting in an outstanding performance on downstream tasks (Gidaris et al., 2018; Noroozi & Favaro, 2016; Doersch et al., 2015; Pathak et al., 2016; Chen et al., 2020a;b; Grill et al., 2020; He et al., 2020).

Despite its effectiveness, most SSL methods require large networks, and the performance deteriorates when the model size is reduced. To address this issue, Fang et al. (2021) proposed SEED, a self-supervised representation distillation method that distills the knowledge of larger pre-trained models into lightweight models via self-supervised learning. Similarly, CompRess (Koohpayegani et al., 2020) mimics the similarity score distribution between a teacher and a student over a dynamically maintained queue. Gao et al. (2022) suggests incorporating consistency constraints between teacher and student embeddings to alleviate the Distilling Bottleneck problem via DisCo. BINGO (Xu et al., 2021) aims to transfer the relationship learned by the teacher to the student by leveraging a set of similar samples constructed by the teacher and grouped within a bag.

Despite achieving state-of-the-art results across multiple tasks with high performance, some concerns remain. First, Gao et al. (2022) discovered that expanding the dimension of the hidden layer in the MLP (projection head) could alleviate the Distilling Bottleneck problem. However, this approach is trivial since determining the size of the dimension and how large it should be remains unanswered. Second, because the student is lightweight with limited capability, it is challenging to accurately mimic the teacher from the encoder to the projection head. For example, in the DisCo study (Gao et al., 2022), they expanded the dimension to 2048, which is the projection head dimension of ResNet-50/101/152. Consequently, this approach is equivalent to increasing the capability of mimicking the teacher, resulting in the improved performance. However, when using ResNet-50$\times$2 with a dimension of 8192 as a teacher, the performance on MobileNet-v3-Large and EfficientNet-B1 drops significantly and is inferior to the previous method (Fang et al., 2021). Moreover, the feature distributions of the teacher and student models are statistically different and cannot be directly compared in practice, even if their dimensions are the same. Therefore, the optimal dimension for the projection head and how to efficiently distill the teacher embedding remain unanswered questions.

In this study, we propose a novel approach for improving the performance of distilling lightweight models through SSL. Specifically, we suggest reusing the pre-trained teacher projection head for students, instead of mimicking it during training. This is based on the hypothesis that the most valuable knowledge is contained in the projection head, and it should be retained during distillation. Our proposed "teacher projection head reusing strategy" involves replacing the student projection head with the pre-trained one from a teacher, which is a large dimension MLP layer that has been optimized. This enables direct reuse of the projection head, without the need for heuristic selection of the dimension size via trial and error. Additionally, a "dimension adapter" is inserted between the student encoder and the teacher projection head to align the dimension.

Our approach simplifies the training objective from mimicking the representation and the embedding to aligning the representation with the optimal embedding. Our experiments show that the proposed method, named RETRO, outperforms the existing DisCo method by a significant margin when using the same architecture on various downstream tasks. Moreover, RETRO

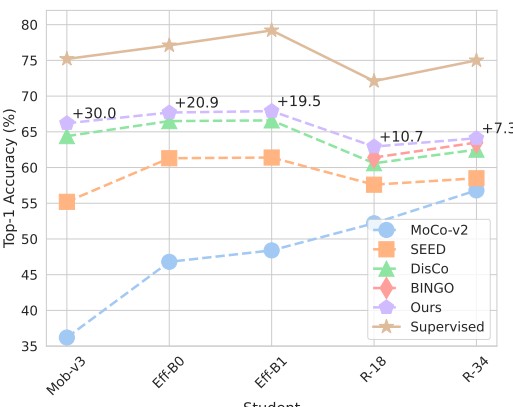

Figure 1: ImageNet top-1 linear evaluation accuracy on different network architectures. Our method significantly exceeds the result of using MoCo-V2 directly and surpasses the state-of-the-art DisCo by a large margin. Particularly, the result of EfficientNet-B0 is quite close to the teacher ResNet-50, while the number of parameters of EfficientNet-B0 is only 16.3% of ResNet-50. The improvement brought by RETRO is compared to the MoCo-V2 baseline.

achieves state-of-the-art SSL results on all lightweight models, including ResNet-18/34, EfficientNet-B0/B1, and MobileNetV3. Notably, the linear evaluation results of EfficientNet-B0 on ImageNet are comparable to ResNet-50/ResNet-101, despite having only a fraction of the parameters. On the COCO and PASCAL VOC datasets, RETRO also achieves more than 3% mAP improvement across different pre-trained models.

## 2 RELATED WORK

Self-supervised learning and knowledge distillation have emerged as crucial research areas in machine learning, attracting significant attention in recent years. In this section, we present a review of some of the key works in these fields.

### 2.1 SELF-SUPERVISED LEARNING

Self-supervised learning is an essential branch of unsupervised learning that automatically generates supervisory signals from unlabeled data. One of the earliest and most effective techniques used in self-supervised learning is the autoencoder, which compresses the input data and reconstructs it. Contrastive learning is another popular self-supervised learning method that enables the model to differentiate between similar and dissimilar pairs of examples.

Recent studies have demonstrated the efficacy of contrastive-based techniques in self-supervised representation learning, where different perspectives of the same input are encouraged to be closer in feature space (Chen et al., 2020a;b; Chen & He, 2021; Chen et al., 2020c; Grill et al., 2020; He et al., 2020; Henaff, 2020; Wang et al., 2021a;b; Zbontar et al., 2021). For instance, SimCLR (Chen et al., 2020a;b) has proved that using strong data augmentation, larger batch sizes of negative samples, and including a projection head (MLP) after global average pooling can boost self-supervised learning. However, the performance of SimCLR is dependent on very large batch sizes and may not be feasible in real-world scenarios.

MoCo (Chen et al., 2020c; He et al., 2020), on the other hand, uses a memory bank to maintain consistent representations of negative samples. It considers contrastive learning as a look-up dictio-

nary, enabling it to achieve superior performance without large batch sizes, making it more practical. BYOL (Grill et al., 2020) introduces a predictor to one branch of the network to prevent trivial solutions and break the symmetry. DINO (Caron et al., 2021) applies contrastive learning to vision transformers with self-distillation intuition.

## 2.2 KNOWLEDGE DISTILLATION

Knowledge distillation is a powerful technique used for transferring knowledge from a large, complex model (known as the teacher) to a smaller, simpler model (known as the student) to improve its performance.

The idea of knowledge distillation was first proposed by Hinton et al. (2015), which transfers knowledge from a large teacher to a smaller student by minimizing the Kulback-Leibler (KL) divergence between the outputs of the two models. Zagoruyko & Komodakis (2016) introduced Attention Transfer (AT) to transfer the spatial attention of the teacher to the student by minimizing the mean squared error (MSE) between the feature maps of the two models. This method guides the student to focus on relevant regions of the input image, improving its performance on small datasets.

FitNets (Romero et al., 2014) is another method of knowledge distillation that transfers knowledge from the intermediate layers of a deep and thin teacher to a deeper but thinner student. The intermediate layers learned by the teacher are treated as hints, and the student is trained to mimic them using mean squared error loss. Relation Knowledge Distillation (RKD) (Park et al., 2019) is a method that transfers the mutual relationship between the samples in a batch from the teacher to the student. RKD uses distance-wise and angle-wise distillation loss to transfer the relationship between the samples to the student.

## 2.3 SELF-SUPERVISED LEARNING AND KNOWLEDGE DISTILLATION

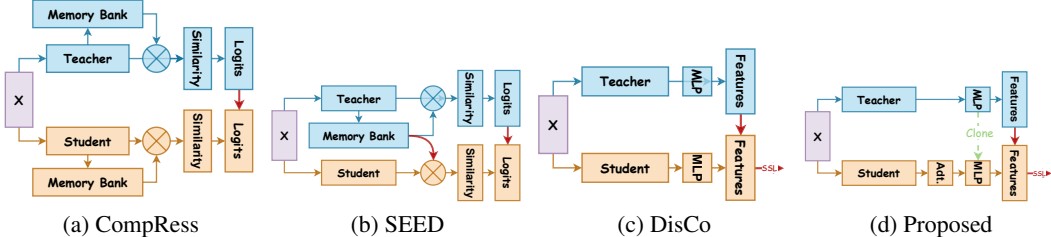

| (a) CompRess | (b) SEED | (c) DisCo | (d) Proposed |

Figure 2: Comparison with existing self-supervised distillers. $x$ is the input image. The orange arrow indicates the knowledge transfer direction. Both 2a CompRess (Koohpayegani et al., 2020) and 2b SEED (Fang et al., 2021) transfer the knowledge of the similarity between a sample and a negative memory bank. 2c DisCo (Gao et al., 2022) constrains the last embedding of the student to be consistent with that of the teacher. 2d Our RETRO improves DisCo by reusing the teacher projection head for the student, which has a higher capability to generate generalized embedding. 'Adt.' indicates the adapter layer.

In recent years, there has been a growing interest in combining knowledge distillation and self-supervised learning to improve the learning process. Some recent works, such as CRD (Tian et al., 2019) and SSKD (Xu et al., 2020), have used self-supervision as an auxiliary task to enhance knowledge distillation in fully supervised settings by transferring relationships between different modalities or mimicking transformed data and self-supervision tasks.

On the other hand, CompRess (Koohpayegani et al., 2020) and SEED (Fang et al., 2021) have focused on improving self-supervised visual representation learning on small models by incorporating knowledge distillation. They leverage the memory bank of MoCo (He et al., 2020) to maintain the consistency of the student's distribution with that of the teacher. Meanwhile, DisCo (Gao et al., 2022) proposes to align the final embedding of the lightweight student with that of the teacher, exploiting the student's learning ability to maximize knowledge. They also increase the dimension of the student's projection head to better mimic the teacher's embedding.

However, the questions of which knowledge is essential for the student and how to efficiently distill it remain unanswered. Moreover, previous approaches focused only on making the student mimic the teacher instead of exploiting the student's learning ability. Our proposed method aims to enhance the self-supervised representation learning ability of lightweight models by aligning the student encoder with the teacher's projection head instead of merely mimicking the teacher. Figure 2 illustrates the differences between our proposed method and CompRess, SEED, and DisCo.

## 3 METHOD

In this section, we will provide a detailed description of our proposed method, RETRO. We will start by reviewing the preliminary concepts of contrastive-learning-based SSL. Next, we will discuss the overall framework of RETRO and explain how it works. Finally, we will introduce the objective of RETRO and describe the process of updating its parameters.

### 3.1 PRELIMINARY ON CONTRASTIVE LEARNING BASED SSL

#### 3.1.1 CONTRASTIVE LEARNING BASED SSL

In contrastive-learning-based SSL, the goal is to predict whether a pair of instances belong to the same class or different classes. The two instances are obtained by applying different data augmentation techniques to the same input image $x$, resulting in two augmented views $v$ and $v'$ of the same instance. The objective is to make the two views similar while views of different instances should be dissimilar. Each view is then passed through two encoders $f_s(\cdot)$ and $f_t(\cdot)$ to obtain the corresponding representations $z_s$ and $z_t$.

To map the high-dimensional representations to a lower-dimensional embedding, a projection head $g(\cdot)$, which is a non-linear MLP, is used. Specifically, $g$ takes the representation $z$ as input and maps it to an embedding $E$ as $E = g(z) = g \circ f(x)$. These embeddings are then used to estimate the similarity in contrastive learning. The projection head is crucial to the success of self-supervised learning, as demonstrated in prior works such as MoCo (He et al., 2020) and SimCLR (Chen et al., 2020a). The encoder can be any network architecture, such as ResNet or EfficientNet.

The projection head consists of two linear layers followed by a non-linear activation function such as ReLU. The output dimension of the projection head is typically set to a smaller value, such as 128, to obtain a low-dimensional embedding.

#### 3.1.2 DISCO

In DisCo, the input $x$ is transformed into two views $v$ and $v'$ using two different augmentation strategies in each iteration. The views are fed into both the student encoder $f_s$ and the teacher encoder $f_t$ to create four representations $z_s$, $z_s'$, $z_t$, and $z_t'$. These representations are then projected using two different projection heads $g_s$ and $g_t$ to produce low-dimensional embeddings $E_s$, $E_s'$, $E_t$, and $E_t'$, respectively. The same process is also applied with the mean student, resulting in representations $z_m$, $z_m'$ and embeddings $E_m$, $E_m'$. The embeddings are then used to compute the contrastive learning loss using InfoNCE loss (Oord et al., 2018), similar to MoCo (He et al., 2020), as follows:

$$\mathcal{L}_{\text{con}} = -\log \frac{\exp\left(\mathbf{q} \cdot \mathbf{k}^+ / \tau\right)}{\sum i = 0^K \exp\left(\mathbf{q} \cdot \mathbf{k}_i / \tau\right)}, \tag{1}$$

where $\mathbf{q}$ is the embedding $E_s$ of the student on view $v$, $\mathbf{k}$ is the embedding $E_m'$ of the mean student on view $v'$, $\tau$ is the temperature, and $K$ is the size of the memory bank. Additionally, the embeddings are used to compute a consistency loss using cosine similarity, which is represented using mean squared error (MSE) as follows:

$$\mathcal{L}_{\text{dis}} = \|E_s - E_t\|_2^2 + \|E_s' - E_t'\|_2^2 \tag{2}$$

### 3.2 RETRO

The overall framework of RETRO is illustrated in Figure 3. RETRO comprises a lightweight student $s(\cdot)$, a mean student $m(\cdot)$, and a pre-trained frozen teacher $t(\cdot)$, which is similar to the DisCo

framework (Gao et al., 2022). However, unlike DisCo, we propose that the pre-trained teacher projection head can be used directly for the students since it contains the most valuable knowledge. Therefore, the objective is to train the student encoder to align the representation with the teacher projection head, instead of learning to mimic the teacher's behavior. In addition, we leverage the power of the multi-view strategy by inputting both views into the mean student and enforcing the similarity constraint on pairs of embeddings between the student and the mean student.

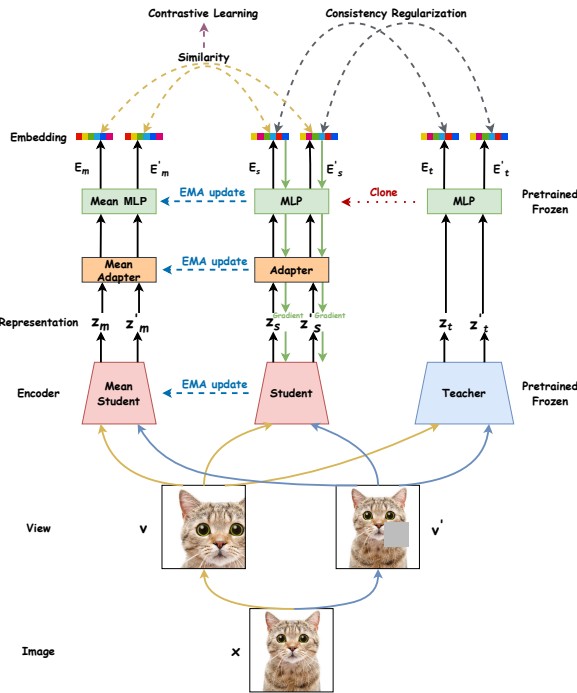

Figure 3: The pipeline of the proposed RETRO technique. Two different data augmentation techniques first transform a single image into two views. A self-supervised pre-trained teacher is added in addition to the original contrastive SSL component, and the final embeddings generated by the learnable student and the frozen teacher must be consistent for each view. The contrastive is conducted with both views, and the projection head is frozen and reused in the student model.

To achieve this, we replace the student projection head with the teacher's projection head $g$, ensuring the consistency of all projection heads. However, since the input dimension of the student projection head is smaller than that of the teacher, we place an adapter $a(\cdot)$ between the encoder and projection head to align the dimension. This process can be formulated as $E_s = g \circ a(z_s)$, $E'_s = g \circ a(z'_s)$, $E_t = g(z_t)$, $E'_t = g(z'_t)$, $E_m = g' \circ a'(z_m)$, and $E'_m = g' \circ a'(z'_m)$, where $a'(\cdot)$ and $g'(\cdot)$ are the mean adapter and mean projection head, respectively. These embeddings are then used to compute the contrastive loss and consistency loss.

### 3.3 LOSS FUNCTION AND PARAMETER UPDATE PROCESS

We follow BYOL (Grill et al., 2020) to symmetrize the contrastive loss in RETRO as follows:

$$\mathcal{L}_{\text{con}} = \frac{1}{2}\left(-\log\frac{\exp\left(\mathbf{q}\cdot\mathbf{k}'_+/\tau\right)}{\sum_{i=0}^{K}\exp\left(\mathbf{q}\cdot\mathbf{k}'_i/\tau\right)}\right) + \frac{1}{2}\left(-\log\frac{\exp\left(\mathbf{q}'\cdot\mathbf{k}_+/\tau\right)}{\sum_{i=0}^{K}\exp\left(\mathbf{q}'\cdot\mathbf{k}_i/\tau\right)}\right) \quad (3)$$

Here, $q$ and $q'$ are the embeddings from the student, while $k$ and $k'$ are the embeddings from the mean student. We use two different memory banks for the two different views $v$ and $v'$, respectively.

The overall loss function of RETRO is formulated as follows:

$$\mathcal{L} = \mathcal{L}_{\text{dis}} + \gamma\mathcal{L}_{\text{con}} \quad (4)$$

where $\mathcal{L}_{\text{dis}}$ is the consistency loss, and $\mathcal{L}_{\text{con}}$ is the contrastive loss of the conventional SSL method. $\gamma$ is a hyperparameter used to control the weight of the contrastive loss, which is typically set to 1. The parameters of the student encoder are optimized using the objective from Equation 4, while the parameters of the entire mean student are updated using the exponential moving average strategy as follows:

$$\theta_k \leftarrow m\theta_k + (1 - m)\,\theta_q \tag{5}$$

Here, $m \in [0, 1)$ is the momentum coefficient, which is typically set to $0.999$, and $\theta$ represents the model parameters.

## 4 EXPERIMENTS

### 4.1 IMPLEMENTATION DETAILS

We first pre-train the self-supervised teacher models on the ImageNet dataset (Russakovsky et al., 2015), which contains 1.3 million training images and $50,000$ validation images with $1,000$ categories, using the MoCo-V2 (He et al., 2020) framework. We use ResNet as the backbone for the teacher models with different widths/depths, such as ResNet-50 (22.4M), ResNet-101 (40.5M), ResNet-152 (55.4M), and ResNet-50×2 (94M). We pre-train ResNet-50/101 using the MoCo-V2 framework for 200 epochs, ResNet-152 for 800 epochs, while ResNet-50×2 is pre-trained using the SwAV (Caron et al., 2020) method for 400 epochs.

We evaluate our RETRO method on five lightweight networks as students, including EfficientNet-B0 (4.0M), EfficientNet-B1 (6.4M), MobileNet-v3-Large (4.2M), ResNet-18 (10.7M), and ResNet-34 (20.4M). We use the same learning rate and optimizer as MoCo-V2 and train all student models for 200 epochs. During distillation, the teacher's and student's projection heads are frozen for RETRO, while SEED, DisCo, and BINGO only freeze the teacher. As a result, RETRO has fewer trainable parameters and simpler training objectives. Note that SEED trains the models using a SwAV pre-trained teacher for 400 epochs, and BINGO uses CutMix regularization (Yun et al., 2019) and more positive samples (5×) during training, resulting in a higher benchmark score. Additionally, BINGO is not an end-to-end framework.

We later fine-tune the self-supervised distillation models for linear evaluation on ImageNet for 100 epochs. We set the initial learning rate to 3 for EfficientNet-B0/B1 and MobileNet-v3-Large, and 30 for ResNet-18/34. The learning rate is scheduled to decrease by a factor of 10 at 60 and 80 epochs, and we use SGD as the optimizer. We follow the other hyperparameters strictly as in MoCo-V2 (He et al., 2020).

### 4.2 LINEAR EVALUATION

The results presented in Table 1 demonstrate that students distilled by RETRO outperform their counterparts pre-trained by MoCo-V2 (Baseline), and also outperform the prior state-of-the-art DisCo by a significant margin. However, we have not included CompRess in our comparison since it uses a teacher that was trained for 600 epochs longer and distills for 400 epochs longer than SEED, DisCo, and RETRO. Therefore, it would be unfair to compare RETRO to CompRess in this context.

The results in Table 1 demonstrate that RETRO outperforms prior methods SEED, DisCo, and BINGO across all benchmarked models. Notably, when using ResNet-50 as the teacher, RETRO achieves state-of-the-art top-1 accuracy on all student models. Moreover, using ResNet-152 instead of ResNet-50 as the teacher leads to a significant improvement in the performance of ResNet-34, from $56.8\%$ to $69.4\%$. It is worth noting that when using RETRO with ResNet-50/101 as the teacher, the linear evaluation result of EfficientNet-B0 is very close to that of the teacher, despite EfficientNet-B0 having only $9.4\%/16.3\%$ of the parameters of ResNet-50/101.

### 4.3 SEMI-SUPERVISED LINEAR EVALUATION

We also evaluate our method in semi-supervised scenarios, following previous methodologies. We use $1\%$ and $10\%$ sampled subsets of the ImageNet training data (i.e., 13 and 128 samples per class, respectively) to fine-tune the student models. As shown in Table 2, our RETRO approach consistently outperforms the baseline under any quantity of labeled data. Notably, our method achieves these

Table 1: ImageNet Test Accuracy (%) using Linear Classification on Different Student Architectures. In the table, ◇ indicates that the teacher and students are pre-trained with MoCo-V2, while † indicates that the teacher is pre-trained by SwAV. SEED distilled for 800 epochs using R-50×2 as the teacher, while DisCo, BINGO, and RETRO distilled for 200 epochs. "T" denotes the teacher, and "S" denotes the student. The subscript in green represents the improvement compared to the MoCo-V2 baseline.

| Method | S / T | Eff-b0 | | Eff-b1 | | Mob-v3 | | R-18 | | R-34 | |
|---|---|---|---|---|---|---|---|---|---|---|---|
| | | T-1 | T-5 | T-1 | T-5 | T-1 | T-5 | T-1 | T-5 | T-1 | T-5 |
| **Supervised** | | 77.1 | 93.3 | 79.2 | 94.4 | 75.2 | - | 72.1 | - | 75.0 | - |
| *Self-supervised* MoCo-V2 (Baseline)◇ | | 46.8 | 72.2 | 48.4 | 73.8 | 36.2 | 62.1 | 52.2 | 77.6 | 56.8 | 81.1 |
| *SSL Distillation* | | | | | | | | | | | |
| SEED (Fang et al., 2021) | R-50 (67.4)◇ | 61.3 | 82.7 | 61.4 | 83.1 | 55.2 | 80.3 | 57.6 | 81.8 | 58.5 | 82.6 |
| DisCo (Gao et al., 2022) | R-50 (67.4)◇ | 66.5 | 87.6 | 66.6 | 87.5 | 64.4 | 86.2 | 60.6 | 83.7 | 62.5 | 85.4 |
| BINGO (Xu et al., 2021) | R-50 (67.4)◇ | - | - | - | - | - | - | 61.4 | 84.3 | 63.5 | 85.7 |
| **RETRO** | R-50 (67.4)◇ | **66.9** | **88.2** | **67.1** | **88.4** | **66.2** | **87.2** | **62.9** | **85.4** | **64.1** | **86.8** |
| | | (20.1 ↑) | (16.0 ↑) | (18.7 ↑) | (14.6 ↑) | (30.0 ↑) | (25.1 ↑) | (10.7 ↑) | (7.8 ↑) | (7.3 ↑) | (5.7 ↑) |
| SEED (Fang et al., 2021) | R-101 (70.3) | 63.0 | 83.8 | 63.4 | 84.6 | 59.9 | 83.5 | 58.9 | 82.5 | 61.6 | 84.9 |
| DisCo (Gao et al., 2022) | R-101 (69.1)◇ | 68.9 | 88.9 | 69.0 | 89.1 | 65.7 | 86.7 | 62.3 | 85.1 | 64.4 | 86.5 |
| **RETRO** | R-101 (70.3) | **69.3** | **89.8** | **69.9** | **89.9** | **67.5** | **88.6** | **64.8** | **86.6** | **66.1** | **87.9** |
| | | (22.5 ↑) | (17.6 ↑) | (21.5 ↑) | (16.1 ↑) | (31.3 ↑) | (26.5 ↑) | (12.6 ↑) | (9.0 ↑) | (9.3 ↑) | (6.8 ↑) |
| SEED (Fang et al., 2021) | R-152 (74.2) | 65.3 | 86.0 | 67.3 | 86.9 | 61.4 | 84.6 | 59.5 | 83.3 | 62.7 | 85.8 |
| DisCo (Gao et al., 2022) | R-152 (74.1)◇ | 67.8 | 87.0 | 73.1 | 91.2 | 63.7 | 84.9 | 65.5 | 86.7 | 68.1 | 88.6 |
| BINGO (Xu et al., 2021) | R-152 (74.1)◇ | - | - | - | - | - | - | 65.9 | 87.1 | 69.1 | 88.9 |
| **RETRO** | R-152 (74.1)◇ | **69.8** | **90.2** | **73.7** | **91.4** | **68.0** | **86.2** | **66.9** | **88.1** | **69.4** | **89.9** |
| | | (23.0 ↑) | (18.0 ↑) | (25.3 ↑) | (17.6 ↑) | (31.8 ↑) | (24.1 ↑) | (14.7 ↑) | (10.5 ↑) | (12.6 ↑) | (8.8 ↑) |
| SEED (Fang et al., 2021) | R-50×2 (77.3)† | 67.6 | 87.4 | 68.0 | 87.6 | 68.2 | 88.2 | 63.0 | 84.9 | 65.7 | 86.8 |
| DisCo (Gao et al., 2022) | R-50×2 (77.3)† | 69.1 | 88.9 | 64.0 | 84.6 | 58.9 | 81.4 | 65.2 | 86.8 | 67.6 | 88.6 |
| BINGO (Xu et al., 2021) | R-50×2 (77.3)† | - | - | - | - | - | - | 65.5 | 87.0 | 68.9 | 89.0 |
| **RETRO** | R-50×2 (77.3)† | **70.2** | **90.4** | **73.8** | **91.6** | **70.1** | **89.2** | **65.9** | **87.1** | **68.9** | **89.7** |
| | | (23.4 ↑) | (18.2 ↑) | (25.4 ↑) | (17.8 ↑) | (33.9 ↑) | (27.1 ↑) | (13.7 ↑) | (9.5 ↑) | (12.1 ↑) | (8.6 ↑) |

results while strictly following the settings from SEED (Fang et al., 2021) and DisCo (Gao et al., 2022), whereas BINGO uses a higher learning rate (10) for the classifier layer.

Moreover, our experiments demonstrate that RETRO is stable under varying percentages of annotations, indicating that students always benefit from being distilled by larger teacher models. The results also suggest that having more labeled data can help improve the final performance of the student models.

Table 2: Semi-supervised learning by fine-tuning 1% and 10% images on ImageNet using ResNet-18.

| Method | T | 1% labels | 10% labels |
|---|---|---|---|
| MoCo-V2 (Baseline) | - | 30.9 | 45.8 |
| SEED(Fang et al., 2021) | R-50 (67.4) | 39.1 | 50.2 |
| DisCo(Gao et al., 2022) | R-50 (67.4) | 39.2 | 50.1 |
| BINGO(Xu et al., 2021) | R-50 (67.4) | 42.8 | 57.5 |
| **RETRO** | R-50 (67.4) | **43.1** | **57.9** |
| SEED(Fang et al., 2021) | R-101 (70.3) | 41.4 | 54.8 |
| DisCo(Gao et al., 2022) | R-101 (69.1) | 47.8 | 54.7 |
| **RETRO** | R-101 (70.3) | **49.2** | **60.5** |
| SEED(Fang et al., 2021) | R-152 (74.1) | 44.3 | 54.8 |
| DisCo(Gao et al., 2022) | R-152 (74.1) | 47.1 | 54.7 |
| BINGO(Xu et al., 2021) | R-152 (74.1) | 50.3 | 61.2 |
| **RETRO** | R-152 (74.1) | **50.9** | **62.0** |
| BINGO(Xu et al., 2021) | R-50×2 (77.3) | 48.2 | 60.2 |
| **RETRO** | R-50×2 (77.3) | **50.6** | **61.9** |

## 4.4 TRANSFER TO CIFAR-10/CIFAR-100

We conducted further evaluations to assess the generalization of representations obtained by RETRO on CIFAR-10 and CIFAR-100 datasets, using ResNet-18/EfficientNet-B0 as a student and ResNet-50/ResNet-101/ResNet-152 as a teacher. The models were fine-tuned for 100 epochs, with an initial

learning rate of 3 and the learning rate scheduler decreasing by a factor of 10 at 60 and 80 epochs. All images were resized to $224 \times 224$, following the methodology from (Fang et al., 2021) since the original image resolution of the CIFAR dataset is $32 \times 32$. The results presented in Figure 4 show that RETRO outperforms prior methods SEED and DisCo across the datasets. Furthermore, the improvement brought by RETRO becomes more apparent as the quality of the teacher models improves.

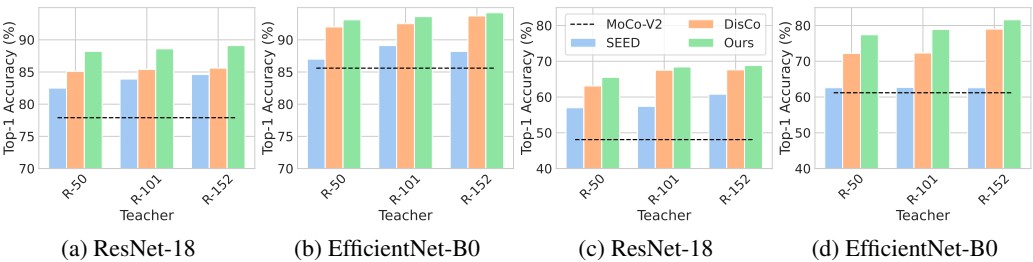

|         |         |         |         |
| :-----: | :-----: | :-----: | :-----: |
| (a) ResNet-18 | (b) EfficientNet-B0 | (c) ResNet-18 | (d) EfficientNet-B0 |

Figure 4: Top-1 accuracy for transfer learning on CIFAR-10 (4a, 4b) and CIFAR-100 (4c, 4d) dataset.

## 4.5 TRANSFER TO DETECTION AND SEGMENTATION

For segmentation and downstream detection tasks, we adopt the fine-tuning methodology used in SEED (Fang et al., 2021) and DisCo (Gao et al., 2022), where all parameters are fine-tuned. For the detection task on VOC (Everingham et al., 2015), the model is initialized with an initial learning rate of 0.1, with 200 warm-up iterations, and the learning rate is decreased by a factor of 10 at 18k and 22.2k steps. The detector is trained for $48k$ steps, with a total batch size of 32. During training, the input is randomly scaled from 400 to 800, and set to 800 during inference. For detection and segmentation on COCO (Lin et al., 2014), the model is trained for $180k$ iterations with an initial learning rate of 0.11. During training, the image scale is randomly sampled from 600 to 800.

Table 3: Object detection and instance segmentation results on VOC-07 test and COCO val2017 using ResNet-34 as the backbone. The subscript in green represents the improvement compared to the MoCo-V2 baseline.

| S | T | Method | Object Detection | | | | | | Instance Segmentation | | |
|---|---|---|---|---|---|---|---|---|---|---|---|
| | | | VOC | | | COCO | | | COCO | | |
| | | | $AP^{bb}$ | $AP^{bb}_{50}$ | $AP^{bb}_{75}$ | $AP^{bb}$ | $AP^{bb}_{50}$ | $AP^{bb}_{75}$ | $AP^{mk}$ | $AP^{mk}_{50}$ | $AP^{mk}_{75}$ |
| | $\times$ | MoCo-V2 | 53.6 | 79.1 | 58.7 | 38.1 | 56.8 | 40.7 | 33.0 | 53.2 | 35.3 |
| R-34 | R-50 | SEED (Fang et al., 2021) | 53.7 | 79.4 | 59.2 | 38.4 | 57.0 | 41.0 | 33.3 | 53.2 | 35.3 |
| | | DisCo (Gao et al., 2022) | 56.5 | 80.6 | 62.5 | 40.0 | 59.1 | 43.4 | 34.9 | 56.3 | 37.1 |
| | | **RETRO** | **57.2** | **81.4** | **63.3** | **41.5** | **60.2** | **45.3** | **35.9** | **57.8** | **38.8** |
| | | | (3.6 ↑) | (2.3 ↑) | (4.6 ↑) | (3.4 ↑) | (3.4 ↑) | (4.6 ↑) | (2.9 ↑) | (4.6 ↑) | (3.5 ↑) |
| | R-101 | SEED (Fang et al., 2021) | 54.1 | 79.8 | 59.1 | 38.5 | 57.3 | 41.4 | 33.6 | 54.1 | 35.6 |
| | | DisCo (Gao et al., 2022) | 56.1 | 80.3 | 61.8 | 40.0 | 59.1 | 43.2 | 34.7 | 55.9 | 37.4 |
| | | **RETRO** | **57.3** | **81.8** | **63.5** | **41.5** | **60.3** | **45.4** | **36.0** | **57.8** | **38.9** |
| | | | (3.7 ↑) | (2.7 ↑) | (4.8 ↑) | (3.4 ↑) | (3.5 ↑) | (4.7 ↑) | (3.0 ↑) | (4.6 ↑) | (3.6 ↑) |
| | R-152 | SEED (Fang et al., 2021) | 54.4 | 80.1 | 59.9 | 38.4 | 57.0 | 41.0 | 33.3 | 53.7 | 35.3 |
| | | DisCo (Gao et al., 2022) | 56.6 | 80.8 | 63.4 | 39.4 | 58.7 | 42.7 | 34.4 | 55.4 | 36.7 |
| | | BINGO (Xu et al., 2021) | - | - | - | 39.9 | 59.4 | 43.5 | 35.7 | 56.5 | 38.2 |
| | | **RETRO** | **57.5** | **81.9** | **64.1** | **41.4** | **60.6** | **45.4** | **36.1** | **57.3** | **39.2** |
| | | | (3.9 ↑) | (2.8 ↑) | (5.4 ↑) | (3.3 ↑) | (3.8 ↑) | (4.7 ↑) | (3.1 ↑) | (4.1 ↑) | (3.8 ↑) |

We also performed tests on detection and segmentation tasks for generalization analysis. Faster R-CNN (Ren et al., 2015) based on C4 is used for object detection for VOC and R-CNN Mask (He et al., 2017) is used for object detection and version segmentation for COCO. The results are displayed in Table 3. In object detection, our method can yield clear improvements for both VOC and COCO datasets. Also, as claimed by SEED (Fang et al., 2021), the COCO training dataset has 118k images, while VOC has only 16.5k training images, so the improvement of COCO is relatively small compared to VOC. Therefore, the gain from initialization weight is relatively small. RETRO also has an advantage when it comes to instance segmentation tasks.

## 4.6 ABLATION STUDY

**Impact of each contribution:** In this section, we report the effectiveness of each of our contributions. ① Reusing teacher projection head and ② symmetric contrastive learning loss. We verify this via students that are trained with ResNet-50 as a teacher.

Table 4: ImageNet top-1 accuracy (%) using linear classification on different strategies.

| Setting / Student Model | Eff-b0 | Eff-b1 | Mob-v3 | R-18 | R-34 |
|---|---|---|---|---|---|
| MoCo-V2 (Baseline) | 46.8 | 48.4 | 36.2 | 52.2 | 56.8 |
| DisCo (Gao et al., 2022) | 66.5 | 66.6 | 64.4 | 60.6 | 62.5 |
| + ① | 66.7 | 66.9 | 65.8 | 62.5 | 63.4 |
| + ② (RETRO) | **66.9** | **67.1** | **66.2** | **62.9** | **64.1** |

**Computational Complexity:** As illustrated in Figure 3, the computational cost of RETRO is higher compared to SEED (Fang et al., 2021) and DisCo (Gao et al., 2022) due to the additional forward propagation required for the mean student. The total number of forward propagation is 6, which is three times higher than SEED and MoCo-V2. However, these additional forward propagations are not used during inference, so there is no overhead at inference time. The results in Table 5 show that RETRO has a lower number of learnable parameters than DisCo. Therefore, the run-time overhead of RETRO is small and negligible compared to DisCo and BINGO. It should be noted that BINGO requires a KNN run to create a bag of positive samples, while RETRO is an end-to-end approach.

Table 5: Comparison for the number of learnable parameters between DisCo and RETRO.

| Method | Eff-b0 | Eff-b1 | Mob-v3 | R-18 | R-34 |
|---|---|---|---|---|---|
| DisCo (Gao et al., 2022) | 6.57M | 8.96M | 6.76M | 11.91M | 21.55M |
| **RETRO** | 6.32M | 8.71M | 6.51M | 11.66M | 21.30M |
| | (↓ 0.25M) | (↓ 0.25M) | (↓ 0.25M) | (↓ 0.25M) | (↓ 0.25M) |

**Comparison with other Distillation:** For further verifying the strengths of RETRO, we conducted the comparison against several different distillation strategies. We include feature-based distillation (KD) and relation-based distillation (RKD), following DisCo (Gao et al., 2022) and BINGO (Xu et al., 2021). As shown in Table 6, RETRO shows superior performance compared with other distillation methods and surpasses them by a large margin.

Table 6: Top-1 linear classification accuracy on ImageNet utilizing various distillation techniques on the ResNet-18 student model (ResNet-50 is used as teacher model).

| Method | Top-1 |
|---|---|
| MoCo-V2 (Baseline) (He et al., 2020) | 52.2 |
| MoCo-V2 + KD (Fang et al., 2021) | 55.3 |
| MoCo-V2 + RKD (Park et al., 2019) | 61.6 |
| DisCo + KD (Gao et al., 2022) | 60.6 |
| DisCo + RKD (Gao et al., 2022) | 60.6 |
| BINGO (Xu et al., 2021) | 61.4 |
| **RETRO** | **62.9** |

## 5 CONCLUSION

In this paper, we introduce Reusing Teacher Projection head strategy (RETRO), a novel approach for efficiently distilling self-supervised pre-trained teachers on lightweight models. Additionally, we impose symmetry contrastive learning to improve the representation between the student and mean student model. Despite its simplicity, our method outperforms prior methods by a large margin, demonstrating the importance of the projection head in distillation on lightweight models with fewer learnable parameters. RETRO does not introduce any overhead during the inference phase. Our experiments show its superior performance across a range of architectures and tasks.

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
