## A  ADAPTER SETTINGS

We visualize the adapter architecture for different student networks in Figure 5. The adapter for ResNet, EfficientNet, and MobileNet-v3 is a 1-D convolution layer that receives output from the student encoder ($D_s$) and aligns that for the teacher projection head ($D_t$), follows by a batch normalization and a non-linear activation layer. Note that the adapter is placed right before the last pooling layer.

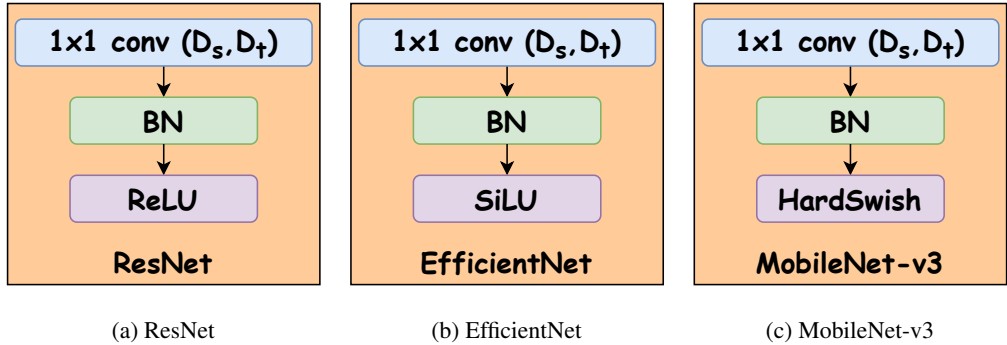

(a) ResNet                      (b) EfficientNet                      (c) MobileNet-v3

Figure 5: Adapter structure for different student networks.

## B  FROZEN VS UNFROZEN PROJECTION HEAD

We conduct the comparison between training RETRO for 200 epochs with frozen projection head and training for 170 epochs with frozen and 30 epochs with unfrozen projection head. As we can see from Table 7, the latter training scheme produces better performance.

Table 7: Comparison for the training scheme of RETRO.

| Method | Eff-b0 | Eff-b1 | Mob-v3 | R-18 | R-34 |
|---|---|---|---|---|---|
| 170 epochs frozen + 30 epochs unfrozen | 66.2 | 66.7 | 65.8 | 62.1 | 63.8 |
| 200 epochs frozen | 66.9 | 67.1 | 66.2 | 62.9 | 64.1 |

This shows that our hypothesis for training to align with the teacher embeddings is correct.

## C  LIMITATIONS

While showing promising results for efficiently distilling lightweight models via SSL, the proposed method still has some limitations. One of them that can be easily noticed is the increment in the number of forward propagation. This increment introduces the training time overhead to the whole process, although it does not affect the inference time. Another one is the adaptability to other architectures such as ViT-based architectures, we leave this for future research.