# OpenReview forum: "Retro: Reusing teacher projection head for efficient embedding distillation on Lightweight Models via Self-supervised Learning"
_ICLR.cc/2024/Conference — ICLR 2024 Conference Withdrawn Submission_

### Official Review · Reviewer_96a3 · 2023-10-30

**Soundness:** 2 fair
**Presentation:** 3 good
**Contribution:** 2 fair
**Rating:** 5
**Confidence:** 4

**Summary:**

The paper introduces a knowledge distillation method for self-supervised learning on efficient network architectures such as MobileNetV3, EfficientNet-B0. The author proposes a simple technique called, RETRO, that reuses the teacher’s projection head for students during the knowledge distillation and this simple technique achieves significant improvements on self supervised learning results of efficient architectures.

**Strengths:**

- The paper is easy to read and the method is quite simple and can be easily plugged into any self-supervised learning pipeline.
- The problem of improving the self-supervised learning for efficient architectures is important, especially on why these models are harder to train with SSL pretext task.
- Despite the simplicity of the proposed technique, RETRO achieves consistent improvements in all the tasks (linear evaluation as well object detection/segmentation) that the authors present and the results are impressive.

**Weaknesses:**

- Despite its simplicity, the method seems merely an engineering trick which has worked on the knowledge distillation for efficient networks.
- Considering the limited novelty in the approach, I expected the authors would have put more effort in understanding why it works.
The paper also does not touch much on why efficient architectures are harder to perform SSL. Is there any specific reason based on the architectural design of these efficient networks? Generally, in supervised learning these efficient architectures are known to be better.
- Is this specific choice of keeping only the frozen projection head necessary in the student model? What if the consistency is maintained at earlier layers as well as the final embedding of the teacher and student networks. It should work in a similar manner.
Ideally, one would expect an ablation study on what part of the teacher network should be embedded in the student network to make SSL work?
- What if instead of the projection head, first few layers of the teacher network are enforced in the student model.
- Do the authors test the same approach of keeping projection head from SSL performed on one network to another (not necessarily efficient)? Can this save training time on another network architecture?

**Questions:**

Please look at the weaknesses mentioned above.
Is the knowledge distillation in RETRO performed while doing pre-training on teacher model or after the SSL training has been performed on teacher model?

---

### Official Review · Reviewer_qqvw · 2023-11-01

**Soundness:** 3 good
**Presentation:** 3 good
**Contribution:** 2 fair
**Rating:** 5
**Confidence:** 4

**Summary:**

This paper introduced a method to improve knowledge distillation of light-weight models. They propose to clone the projection head of the teacher to simplify mimicking its preceding features. The method optimizes a contrastive loss between the student and Mean student, and a consistency loss between the student and teacher networks, simultaneously.

**Strengths:**

1) The writing of the main context is clear and well organized.
2) The authors provide thorough explanations of their motivation and methodology, and support their claims with experiments.
3) Quality and reproducibility are acceptable.

**Weaknesses:**

1) The main weakness is the incremental novelty of this work when compared with previous works, particularly DisCo.
2) The method does not consistently lead to significant improved accuracy over baselines in table 1. It is suggested that authors perform some statistical tests to measure if the performance gains are statistically significant and discuss the properties of the architectures and datasets which leads to performance improvement.
3) The computational cost of the method is higher compared to the baselines such as SEED and DisCo.
4) The method is limited to CNN architecture.
5) The related work is lacking KD publications from 2023 such as [1].

[1] Song, Kaiyou, et al. "Multi-Mode Online Knowledge Distillation for Self-Supervised Visual Representation Learning." Proceedings of the IEEE/CVF Conference on Computer Vision and Pattern Recognition. 2023.

**Questions:**

This work can be improved if the authors address my concerns above.

Why are the pretraining algorithm of the teachers unidentical in the experiments?

What are the design choices of the adapter layer in figure 5?

There is a typo in equation 1.

---

### Official Review · Reviewer_dnj5 · 2023-11-01

**Soundness:** 3 good
**Presentation:** 2 fair
**Contribution:** 2 fair
**Rating:** 3
**Confidence:** 5

**Summary:**

This paper proposes a self-supervised method for lightweight models. They use a pretrained teacher and use two loss to train the student: 1. SSL loss on the student 2. Reuse teacher’s MLP layer on student and regress teacher’s features. They show that this simple idea outperforms DisCo on ImageNet and transfer learning to other tasks.

**Strengths:**

[+] The proposed method is simple and effective.
[+] Comprehensive comparison to DisCo on ImageNet and transfer learning shows the effectiveness of their method compared to DisCo.

**Weaknesses:**

[-] Although it’s an interesting insight that using teachers MLP improve the student’s representation, this work lacks technical novelty in my opinion.

[-] Teacher model in the experiments are old SSL methods with only CNN architecture. For example, DINO ViT-B has 80% ImageNet Linear Top-1 acc (DINOv2 has 86.3 with ViT-L). Current setup with weak/outdated teacher is not practical and convincing. I highly encourage the authors to evaluate their model on recent SOTA SSL methods to improve the impact of the paper.

[-] Missing citations :

[b] SimReg: Regression as a Simple Yet Effective Tool for Self-supervised Knowledge Distillation. K L Navaneet, Soroush Abbasi Koohpayegani, Ajinkya Tejankar, Hamed Pirsiavash

[a] effective self-supervised pre-training on low-compute networks without distillation. Fuwen Tan, Fatemeh Saleh, Brais Martinez

**Questions:**

-